# Dormancy Versus Germination: 3D Protein Modeling and Evolutionary Analyses Define the Roles of Genetic Variants in the Barley MKK3 Enzyme

**DOI:** 10.3390/ijms27010530

**Published:** 2026-01-05

**Authors:** Maria Hrmova, Christoph Dockter, Flavia Krsticevic, Morten Egevang Jørgensen, Birgitte Skadhauge, Geoffrey B. Fincher

**Affiliations:** 1School of Agriculture, Food and Wine, Waite Research Institute, Adelaide University, Waite Campus, Glen Osmond, SA 5064, Australia; geoffrey.fincher@adelaide.edu.au; 2Carlsberg Research Laboratory, J. C. Jacobsens Gade 4, 1799 Copenhagen V, Denmark; christoph.dockter@carlsberg.com (C.D.); flavia.krsticevic@carlsberg.com (F.K.); morten.jorgensen@carlsberg.com (M.E.J.); birgitte.skadhauge@carlsberg.com (B.S.)

**Keywords:** active site mechanism, evolutionary analysis, kinase-kinase docking, *Hordeum vulgare*, phosphorylation sites, plant mitogen-activated protein kinases, structure-function

## Abstract

Dormancy is a characteristic of plant seeds that has evolved to avoid exposing the young seedling to adverse weather conditions. The mitogen-activated protein kinase MKK3 from barley is known to mediate the duration of dormancy and subsequent germination of the grain. Here, we used computational and phylogenetic approaches to define the structural model of the monomeric MKK3 domain in complex with the downstream MAPK protein kinase that it phosphorylates. We utilized key genetic variants of the barley MKK3 and generated the structural MKK3/MAPK enzyme-substrate complex, supported by evolutionary analyses, to rationalize the effects of the MKK3 variants occurring at the ATP binding site and in the loops that can be phosphorylated. We propose the likely mechanism of ATP hydrolysis and the effects of common genetic variants on MKK3 activity, thereby influencing the duration of dormancy. The data will facilitate future manipulations of dormancy length in different environments.

## 1. Introduction

Dormancy, which is defined as the suppression of grain germination during environmental conditions that would normally be suitable for germination, is a crucial process for the survival of cereals and other plants [1]. In the wild, longer dormancies generally provide a level of insurance by releasing the brakes on dormancy over an extended period, during which the chances of favorable weather conditions are increased. For example, in Mediterranean climates where many cereal crops are grown, grain maturation occurs in early summer. If infrequent rainfall events subsequently occur immediately after grain maturity, a non-dormant grain might germinate, only to die during the ensuing heat of the summer. Dormancy ensures the grain does not germinate until the cooler, wetter weather of the autumn. This is not the case in cooler or wetter climates, where protection against heat and water stress is not necessary and local lines might evolve shorter dormancies. In some cases, a short dormancy allows two crops to be grown in a single year. In a commercial context, where barley is used for malting, brewing and distilling, a moderate length of dormancy is desirable to avoid quality losses due to preharvest sprouting (PHS), where the grain germinates before harvest while still on the spike. However, it is also important for brewing and related industries that the grain can be readily germinated after a short storage period. Thus, the length of dormancy has been an important selection criterion for barley breeders, and the small number of controlling genes has been investigated over many years.

Early QTL analyses revealed two dormancy loci in barley [2]. The dormancy gene at one of these loci was subsequently shown to encode an alanine aminotransferase (AlaAT), which could play a biochemical role in the embryo by linking nitrogen and carbon metabolism with protein synthesis as sugars and amino acids are released from storage proteins and starch during the germination process [3,4]. The dormancy-related gene within the other QTL was shown to encode a mitogen-activated protein kinase (MAPK), specifically the barley mitogen-activated protein kinase kinase 3 (MKK3) [5]. The importance of the MKK3 enzyme was confirmed by Jørgensen et al. [6], who described anthropogenic selection of MKK3 haplotypes post domestication and showed that a mutant barley line with a stop codon in the MKK3 gene was ‘fully dormant’. Mitogens are signal transduction molecules that enhance mitosis, which occurs after germination of the grains when cells in the embryo divide rapidly, expand and elongate as the coleorhiza and coleoptile force their way out of the grain. Mitogen-activated protein kinases in barley and other plants are arranged as a series of enzymes leading from the mitogen receptor, usually at the plasma membrane, through a cascade of kinases that ultimately activates transcription factors in the nucleus (Figure 1). The MAPK cascade involves the progressive phosphorylation and activation of individual enzymes (Figure 1). The transcription factors mediate transcriptional reprogramming, according to environmental or developmental signals [7,8].

The MAPK pathway (Figure 1) is complex and subject to genetic, environmental and physiological regulation, and enzyme nomenclature across species is not standardized [11,12]. MAPKs at each level of the cascade are encoded by multi-gene families [9]. In many cases, details of the activating mitogen, the mitogen receptor and the target transcription factor have not been defined. However, in general terms, the MAPK cascades in plants are components of regulatory networks that relay external signals to the cytoplasm and nucleus, where alterations in protein activity or gene expression patterns trigger appropriate internal responses to the external signal. External signals include abiotic stresses, such as water and heat stress, hypoxia, oxidative stress, salinity and high heavy metal concentrations, and biotic stresses caused by fungal, bacterial and viral attack [9]. The three-component MAPK module can also mediate signaling by phytohormones such as auxins, abscisic acid (ABA), gibberellic acid (GA), ethylene and cytokinins, which lead to developmental changes, the dormancy/germination transition, cell division, and differentiation [9].

For example, an MKK3 from Arabidopsis phosphorylates an ethylene response factor (ERF4) that represses germination, but ERF4 is inactivated by the phosphorylation and dormancy is released [9]. This results in the synthesis of expansins, which are involved in loosening cell walls and hence facilitate cell expansion and elongation in the developing seedling [13]. In rice, the MKK3 kinase may regulate dormancy via ABA deposited during grain maturation [14]. GA, which stimulates the mobilization of storage carbohydrate and protein in grains of grass species after germination [15], has also been linked with the MAPK cascade in wheat, where GA represses the expression of several members of the MAPK cascade [16]. In rice, signal transduction pathways are mediated by ABA and other phytohormones through MAPK cascades [17,18].

The size of each MAPK gene family varies widely across lineages. In barley (*Hordeum vulgare*), a motif- based search of the MorexV3 proteome [19] identified 18 MAPKs, seven MAPKKs, and 206 MAPKKKs, consistent with the pronounced expansion of MAPKKKs commonly observed in plant genomes. These numbers are broadly comparable with those reported in other angiosperms, including *Arabidopsis thaliana* (20 MAPKs, 10 MAPKKs, and 80 MAPKKKs) and rice (*Oryza* spp.: 17 MAPKs, eight MAPKKs and more than 75 MAPKKKs), but they far exceed the family sizes found in animals, where signaling networks are more streamlined. For example, in humans, 14 MAPKs, eight MAPKKs, and approximately 24 MAPKKKs genes are found [20,21,22,23]. Unicellular model organisms such as the green alga *Chlamydomonas reinhardtii* typically contain only two to three MAPKs [24], providing a clear contrast to the extensive diversification of the MAPK family observed in plants.

In a recent study, Jørgensen and co-authors [6] demonstrated that the MKK3 gene copy number in barley is highly variable and correlates with MKK3 transcript abundance. These authors also investigated the effects of genetic variants of MKK3 on enzyme activity and confirmed that genetic variants that increase MKK3 activity are associated with shorter dormancy. Two key genetic variants of the barley MKK3 have been shown to affect MKK3 enzymic activity, namely E165Q [6] and N260T [5]; in both cases, the variant containing the amido groups (Q and N) had higher enzyme activities, which were associated with shorter dormancy but increased risk of PHS. Thus, in domesticated barley, a combination of amino acid residue substitutions in the MKK3 enzyme and gene copy number variation has resulted in the regulation of MKK3 activity and the length of grain dormancy.

Kinase enzymes occur in both phosphorylated and unphosphorylated forms, allowing for interconversions between their active and inactive conformations, respectively [25]. These phosphorylation processes target highly conserved activation segments and can proceed through autophosphorylation or can be catalyzed by other kinases [26]. In the present study, we employed structural computational approaches to define the mechanism of action of the barley MKK3 enzyme, designated MAPKK in Figure 1 (but referred to as HvMKK3 hereafter), at the molecular level. A BLASTP search (accessed on 2 August 2025) of the Protein Data Bank (http://www.rcsb.org/) with HvMKK3 revealed at least 100 kinase 3D structures, mostly of *Homo sapiens* origin, with ≤32% sequence identity; these structures have been solved by X-ray crystallography in both phosphorylated and unphosphorylated forms. The nearest plant structure to the HvMKK3 model from the AlphaFold 3D Structure Database [27,28] is the phosphorylated form of MAPK5 from *Arabidopsis thaliana* (AtMKK5) [29]. This structure, at 3.20 Å resolution (PDB accession 7XBR, chain A), offers unprecedented detail of its SSVGTIAY phosphorylation site and a neighboring conserved DFG activation motif. This structure reveals the molecular mechanisms of the enzyme’s activation and substrate recognition in Arabidopsis. However, the Arabidopsis MKK5 enzyme has 295 residues [20] while the barley MKK3 has 523 residues [6]; the enzymes share only 23% sequence identity. Jørgensen et al. [6] used AlphaFold and AlphaFill resources [27,28] to retrieve the 3D structure prediction of the monomeric barley MKK3 enzyme domain.

Here, we provide further insights into the molecular mechanism of MKK3 catalysis through computational modeling of the binding of the barley MKK3 enzyme to its immediate downstream MAPK substrate and to the phosphoryl donor, ATP. To avoid nomenclature difficulties [11,12], this barley downstream MAPK is hereafter referred to as HvMAPK. We provide a workable hypothesis of how the activated MKK3 and HvMAPK enzymes interact during the phosphorylation of HvMAPK’s activation loop and how key residues encoded by specific MKK3 genetic variants affect activity, and hence the transition from dormancy to germination.

## 2. Results and Discussion

### 2.1. Phylogeny of MAPKs and Codon Selection

The PF00069 kinase domains recovered 206 MAPKKKs in the cultivar Morex barley reference genome, a substantially larger repertoire than previously reported and reflecting the strong expansion of this family in barley [19]. To understand the structure of this expansion, we classified all MAPKKKs according to their diagnostic Prosite motifs, corresponding to the three canonical subfamilies: RAF-like (GTxx[W/Y]MAPE; motif #2), MEKK-like (G[T/S]P×[W/Y/F]MAPEV; motif #1/motif #3 #1), and ZIK-like (GTPEFMAPE[L/V]Y; motif #3).

Mapping these motif classes onto the phylogeny revealed a highly asymmetric distribution across the major clades (Figure 2). Phylogenetic analysis of the MAPKKKs based on PF00069 domains revealed a highly asymmetric family architecture dominated by a large expansion of RAF-like kinases carrying the GTxx(W/Y)MAPE motif. These RAF-like MAPKKKs formed several deeply supported lineages, including three compact subclades and a broad subclade mixed in the tree, together accounting for more than 85% of all barley MAPKKKs. In contrast, MEKK-like and ZIK-like kinases were located in well-defined clusters. This pattern highlighted a strong lineage-specific proliferation of RAF-like MAPKKKs in barley, representing a major expansion relative to MAPKKs and MAPKs (Figure 2) and reflecting dynamic evolutionary pressures acting on the upper tier of the MAPK cascade.

Codon-based selection analyses using HyPhy FEL and MEME on the *Triticeae* MKK3 codon alignment revealed strong and widespread purifying selection across the kinase domain. FEL identified 163 of 386 variable codons as significantly constrained (β < α, *p* ≤ 0.1), whereas only a single codon showed evidence of diversifying positive selection (β > α, *p* ≤ 0.1). The distribution of site-wise *dN:dS* values was strongly skewed toward values below 1, indicating long-term evolutionary constraint on MKK3 across the *Triticeae* (Appendix A). MEME did not detect additional sites under episodic selection, confirming that positive selection was minimal through the evolutionary history of this gene family.

### 2.2. Ancestral Origin of Genetic Variants E165Q and T260N

Variation in grain dormancy is an important strategy for population survival. Given the wide geographic distribution and adaptation of barley to diverse environments, abundant natural variation in MKK3 in wild barley (*Hordeum spontaneum*) complicates the identification of wild-type genetic variants [6,30]. The wild-type of the E165Q and N260T variants can be affected by natural selection (e.g., shorter dormancy in cooler, wetter climates) or by selection intervention of agrarian farmers and barley breeders. Extensive genetic analyses of barley collections and a pangenome panel led Jørgensen et al. [6] to conclude that the ancestral form of E165Q was the longer dormancy E165 variant. Consistent with this conclusion was the presence in the wild barley progenitor of cultivated barley cultivars, *Hordeum spontaneum*, of a single MKK3 gene, which is the E165 variant.

Ancestral sequence reconstruction using the HvMKK3 sequence confirmed these conclusions, where the long dormancy E165 variant is present in the ancestral root (Appendix A) and is therefore likely to be the long dormancy wild-type variant; the Q165 variant was likely selected multiple times [6].

In contrast, the N260T variant appears to have evolved from G260 to N260 from an ancestral root sequence through several predecessors (Appendix A). However, these predecessors do not include T260, although ancestor 299 (magenta circle) has a serine residue (S260) in the equivalent position to N260 (Appendix A). An additional five extant plant MKK3 enzymes (KAA6418469.1, THU52963.1, XP_020269183.1, XP_021892044.1, XP_021892045.1) carry the S260 residue. It is noteworthy that S260 could mutate to T260 via a single-nucleotide polymorphism (SNP). In pioneering work, Nakamura et al. [5] showed that the N260 variant is evolutionarily conserved from green algae to moss and to seed plants and concluded that the N260 haplotype existed in wild barleys that were domesticated in the Fertile Crescent. As the domesticated N260 haplotypes moved eastward, the N260T mutation evolved (probably via a SNP), and the higher dormancy T260 variant became prevalent in the wet regions of East Asia, where PHS risks are high [5]. Similarly, Jørgensen et al. [6] showed that in warm, high rainfall regions that have a high risk of PHS, a single copy of the T260 variant, which had reduced MKK3 activity and associated higher dormancy, was found in most cultivars. They also suggested that the T260 variant may represent a secondary mutation that has evolved multiple times to reintroduce dormancy in regional barley cultivars with low dormancy levels. It seems likely that the G260 ancestor, if it occurred, has been lost.

To further investigate the evolution of the N260T variant, a codon-based selection analysis of MKK3 orthologues across the *Triticeae* was performed. Nucleotide substitutions can be divided into nonsynonymous substitutions (N), which result in amino acid residue changes, and synonymous substitutions (S), which do not cause residue changes. Positive selection is indicated if the *dN:dS* ratio is greater than 1 (where *dN* and *dS* represent the rates of nucleotide substitution), while stationary (stabilizing) or purifying (negative) selection is indicated by *dN:dS* values that approach 0 [31]. At the codon corresponding to residue 260 in barley (alignment position 262), FEL estimated an α = 1.466 and β = 0 (site-wise *dN:dS* = 0, *p* = 0.0635), indicating moderate purifying selection at this position. Notably, the two adjacent codons (259 and 261) exhibit very strong purifying selection, with near-zero *dN:dS* and highly significant *p*-values (*p* ≤ 0.01). These results show that N260 is embedded in a locally highly conserved region of the kinase domain where amino acid residue substitutions are generally disfavored over long evolutionary timescales.

In population data, T260 is not observed in wild barley accessions and is restricted to landraces and cultivated lines [6]. The strong long-term purifying selection at and around N260 across the *Triticeae*, combined with the absence of T260 in wild barley, argues against an origin via natural adaptive selection. Instead, the data are consistent with N260 being a recent, domestication-associated mutation that arose within a constrained site and has been retained under anthropogenic selection.

In summary, these analyses indicate that E165 was the original long dormancy variant, with relatively low levels of MKK3 activity, while N260 is likely to be the wild-type of the N260T variant. The Q165 and N260 variants are associated with higher MKK3 enzyme activity and shorter dormancy, but with an increased risk of PHS (Table 1).

### 2.3. Phosphorylation Sites of MKK3 and MAPK

Protein kinases phosphorylate the hydroxylated amino acid residues of serine (Ser, S), threonine (Thr, T), and tyrosine (Tyr, Y), using ATP as a phosphoryl donor. Sequence alignment with MusiteDeep [32] (accessed on 10 July 2025) predicted a consensus sequence for the phosphorylation loop of MAPKK enzymes of TFVGTVTY (T245-Y252 of MKK3) and that two Thr (T) residues (underlined) in the barley MKK3 enzyme are phosphorylated in this motif. The amino acid sequences of the two enzymes are shown in Appendix A.

The AtMKK5 3D structure, for which the patterns of the SSVGTIAY phosphorylation motif and conserved neighboring DFG activation segment were established by X-ray crystallography [29], superposes on the HvMKK3 model (Figure 3) with a RMSD (root-mean-square deviation) value of 1.03 Å (231 Cα backbone atoms), which is low given that they share the twilight 32%/47% sequence identity/similarity values. In AtMKK5, the phosphorylated SSVGTIAY motif (the first underlined S221 carries the phosphate group) aligns with that of the MusiteDeep-predicted TFVGTVTY motif in HvMKK3 (phosphorylatable Thr residues underlined) [32], suggesting that the 1st Thr in this motif could be phosphorylated (Figure 4).

For the barley MAPK enzyme, the consensus phosphorylation loop is SLKGTPY (S163-Y169), and the predicted phosphorylated residues are the S163 and T167 residues in the SLKGTPY motif (Appendix A). This S163 residue in HvMAPK is highly conserved and is present in all ancestors (Appendix A), as identified by ancestral sequence reconstruction, conducted by FireProt^ASR^ v2.0 [33,34].

### 2.4. Docking of Activated MKK3 with MAPK

Two groups of models of the HvMKK3/HvMAPK complex were calculated, which tested the uncertainty of protein complex formation, as a myriad of potential docking poses could be adopted in conformational space. We used the two independent approaches of the HDOCK (Figure 3) and ClusPro (Figure 4) algorithms. This reciprocal docking included HvMAPK docked into HvMKK3 and vice versa, that is, where HvMKK3 was docked into HvMAPK. Notably, the resultant complexes exhibited similar orientations of HvMAPK in relation to HvMKK3, allowing HvMAPK to be phosphorylated by HvMKK3 (Figure 3 and Figure 4). In these complexes, HvMAPK bound in the cleft of HvMKK3 such that its phosphorylation motif SLKGTPY (S163-Y169) was proximal to the TFVGTVTY motif (T245-Y252; two Thr phosphorylatable residues underlined) of MKK3. The RMSD value for 789 and 797 residues of HDOCK and ClusPro complexes was 1.75 Å, the value for the 523 residue of the HvMKK3 molecule was 0.24 Å, and the value for the 274 and 266 residues of HvMAPK was 2.75 Å, using the PDB Pairwise Structure Alignment tool with the Java Combinatorial Extension (jCE) algorithm [35]. These RMSD values, using two independent approaches, indicate that the two complexes could represent intermediates during complex formation.

When proteins interact, they recognize each other via long-range attractive and at the same time repulsive electrostatic forces that guide their binding based on shape and chemical complementarity. In the later stages, short-range electrostatic, H-bond, and distance-dependent van der Waals and hydrophobic forces keep proteins in close contact. In the HvMKK3/HvMAPK complex (Figure 3 and Figure 4), we observed that the interfacial residues formed electrostatic interactions and H-bonds, stabilized by van der Waals and hydrophobic forces, at separations between 2.5 Å and 3.3 Å. The first complex (Figure 3) utilized electrostatic and H-bonds, while the second complex (Figure 4) was held together mainly by electrostatic forces, reflecting close interactions of both proteins.

Using a rigid body HDOCK approach, we also conducted docking of an ATP molecule into HvMKK3 itself or into the HvMKK3/HvMAPK complex (Figure 5). We found that ATP bound accurately in the active site cavity of HvMKK3 in both cases, with several optimal poses of ATP identified. However, docking HvMAPK into the HvMKK3-ATP binary complex did not generate productive HvMAPK poses. These computational experiments suggest that ATP most likely diffuses into the HvMKK3/HvMAPK complex during its formation or shortly after it has been formed.

### 2.5. All-Atom Structural Dynamics of HvMKK3/HvMAPK Model Complexes

All-atom MD simulations (5000 models clustered into ten trajectories) using CABS-flex 3.0, based on coarse-grained simulations integrated with all-atom reconstruction, revealed deviations in relative partialities from the dispositions of residues in the starting structures [36,37]. These calculations estimated the best possible convergence between simulations and consensus protein fluctuations, using the statistical knowledge-based force field with implicitly considered solvent effects in CABS-flex 3.0 [37,38]. These computationally inexpensive approaches are conceptually based on ‘one bead per residue’ and describe all-atom protein structures [39]. Charted RMSF values or averaged RMSD values over simulation time (reaching convergence after 25 nsec), plotted relative to residue numbers, inform the overall protein dynamics and flexibility through trajectories of structural ensembles (Appendix A).

The RMSF peak values of elements, carrying IAL**E**YMDG/IAL**Q**YMDG motifs (I162–G169; E165 and Q165 in bold) in HvMKK3 (one structural medoid represents 500 models) differed by 2 Å between the wild-type (E165) and Q165 variant complexes, pointing to changes in protein flexibility in the variant (Appendix A; cf. black and blue traces in panel A for HvMKK3). This comparison revealed that the secondary structures in the IAL**Q**YMDG region had higher degrees of flexibility in the Q165 variant than the remainder of its structure, and that this flexibility may be correlated with catalytic activity. Comparisons of the RMSF values of the wild-type (E165) and variant (Q165) complexes (Appendix A; cf. black and blue traces for HvMKK3) uncovered a slightly decreased average value for Q165, while for its nearby coil/H-bonded-turn-localized YMDG residues, these changes were more significant. These changes in flexibility, coupled with those in local distributions of surface electrostatic charges (Appendix A), could be dominant and may affect the catalytic properties of the Q165 variant. This could also change the protein packing and interior distributions of cavities and tunnels, analyzed by MOLEonline [40] and FPocketWeb1.0.1 [41]. These combined effects may collectively increase the catalytic efficiency of HvMKK3 of the Q165 variant.

Conversely, the changes in flexibility of the ERIR**N**EN/ERIR**T**EN structural elements (E256–N262; N260 and T260 in bold), where N260 and T260 reside, were less dramatic (Appendix A; cf. black and red traces for HvMKK3). Here, the T260 variation increased the structural flexibility of the T260-neighbouring E261 residue (with a carboxyl group), and Y253 (with an ionizable OH-group) (Appendix A; red trace for HvMKK3). These changes could be correlated with the decreased HvMKK3 activity in T260, as they directly influence local surface charge distributions (Appendix A). It is notable that the changes in the RMSF values of bound HvMAPK in the T260 variant were more significant than those in the Q165 variant (Appendix A; cf. black, blue, and red traces in panel A for HvMAPK). This T260 residue, located at the edge of the HvMKK3 structure directly neighbors HvMAPK, which could affect the consequent complex formation.

### 2.6. Effects of the E165Q Genetic Variant on MKK3 Structure

As listed in Table 1, this major E165Q genetic variant is associated with changes in MKK3 enzymic activity and hence dormancy. The E165 variant has lower MKK3 activity and longer dormancy; it is most likely the enzyme wild-type form (Table 1) [6]. In Figure 5, we show that the acidic E165 residue hydrogen bonds (H-bonds) with residues E147, K226, and M167 (through the Cα backbone), which are at separations between 2.8 Å and 3.2 Å, and with the adenine moiety of the ATP substrate, making short contacts at 3.5 Å. In contrast, the neutral Q165 variant, which has higher MKK3 activity and lower dormancy (Table 1), forms H-bond contacts with four residues (E147, M167, N220, and K226) at separations between 2.8 Å and 3.1 Å, and binds ATP closer at 3.2 Å. These differences in distances between the E165 and Q165 variants are small, although they can affect the disposition of the ATP molecule. However, two salt bridges (D229–K116 and E165–K226) are located near E165, and could be affected by the E165 substitution into the Q165 amide form. Secondary structure element distribution and long-range separations between Q165 and ATP-binding residues in HvMAPK are also modified in the Q165 variant, but again, this should not affect the catalytic function of the Q165 residue negatively.

However, it is also likely that the differences in MKK3 activity could result from combined changes in electrostatic surface potentials (calculated by the Adaptive Poisson-Boltzmann Solver [42]; Appendix A). There was a weaker negative electrostatic potential distributed around the ATP binding cavity in the Q165 variant, compared to that of E165, due to the replacement of the carboxylic group of Glu by an amide group in Gln (Figure 5). This could lead to a tighter ATP binding by Q165, which could lead to an increase in catalytic activity, meaning that in the E165Q genetic variant, there is an effect at the ATP binding site.

### 2.7. Effects of the N260T Genetic Variant on MKK3 Structure

The T260 variant participates in H-bonds with the R257 and R259 backbone residues at separations between 2.7 Å and 3.1 Å, while the N260 variant participates in H-bonds with the E256 and R257 backbone residues at separations between 3.1 Å and 3.2 Å (Figure 6). This suggests that the T260 residue is more tightly constrained than N260. The T260 residue loses one long-distance interaction to the phosphorylation motif TFVGTVTY in HvMKK3; there are four long-distance interactions in N260 and three in T260. In addition, there is a greater negative surface potential distribution near T260 compared with N260 (Figure 6). These changes could induce conformational variations, affect dissociation states of residues during complex formation, and slow down the phosphorylation process of HvMAPK by HvMKK3. In both cases, these differences would be expected to decrease the MKK3 activity of the T260 variant, as observed (Table 1). Thus, the N260T variant appears to affect the MAPK phosphorylation site.

Examination of the electrostatic potentials of the T260 variant shows that the replacement of N260, which faces bulk solvent at the edge of the structure (Appendix A), with Thr affects the local surface charge distribution of the HvMKK3 molecule. More specifically, ND2 of N260 forms an H-bond with its backbone O atom at a separation of 2.84 Å and makes an immediate partial electrostatic contact. However, upon the N260T mutation, this partial electrostatic stabilization no longer affects the protein structure as the ND2 atom of N260 has been removed. Instead, the OG1 atom of the polar hydroxyl group of the T260 sidechain, also at the separation of 2.84 Å (cyan), forms an H-bond with its oxygen O atom. Thus, the mechanistic consequences of the T260 variation on local and global surface electrostatic potentials can be rationalized (Appendix A), although we must avoid over-interpretations of the consequences of mutations because we work with protein complex models generated by docking, and the accuracy of these complexes has not yet been backed by electron densities or cryogenic-electron-microscopy maps.

### 2.8. Effects of Other Common MKK3 Variants

Although most attention has been focused on the effects of the E165Q and T260N genetic variants of the barley MKK3 enzyme and their effects on enzyme activity and dormancy, several other variants are known, including A79V, G350R, and N383D [5,6]. All these variants could have arisen from SNPs. Nakamura et al. [5] noted that the G350R and D383N variants are found in both dormant and non-dormant varieties. Recent pangenome analyses highlighted that both variations can be found in single- or multi-copy haplotypes and in combination with other SNPs. Each of these has the potential to further influence kinase function, thus obscuring individual SNP effects on grain dormancy.

In Figure 7, the positions of these additional variants are shown to predict whether the changes are likely to alter MKK3 activity. The V79 residue of the A79V variant is positioned on the enzyme surface, about 29 Å from the catalytic residues, although the large size of the hydrophobic Val side chain might cause changes in the overall structure of the enzyme, particularly in the region of the ATP binding cavity.

The G350R variant residue is also on the surface of the protein, with the charged Arg residue (R) projecting out from the enzyme’s surface. These residues, although a long way (about 30 Å) from the active site, might still influence overall folding in their immediate vicinity. Finally, the N383 variant has been associated with mild increases in MKK3 activity [6], but is situated 35 Å from the active site. These rather large separations between G350R and N383D residues and the active site cavity and phosphorylation segments suggest that their impact on the activity of HvMKK3 is likely minimal. As noted above, the higher activity of the V79 variant may be attributable to changes in the overall 3D structure of the enzyme (Table 1).

### 2.9. Mechanism of MKK3 Action: A Working Model

The activated MKK3 could be phosphorylated at one or two of the Thr residues in its TFVGTVTY (phosphorylatable Thr residues underlined) motif and binds, or retains, a previously diffused ATP molecule. Positively charged residues K116 and K177, together with possible contributions from D211 and D229, bind to the ATP reaction substrate (Figure 3, left panel). The adenine moiety of the ATP is bound near E165Q in this genetic variant. The radius of gyration of ATP is about 14 Å; this places the γ-phosphate atom of bound ATP 19–24 Å from the target phosphorylation site of S163 in the SLKGTPY (phosphorylatable Ser residue is underlined) motif of the HvMAPK enzyme (Figure 3 and Figure 4). This is too far for direct phosphoryl transfer, which is observed in other protein kinase enzymes through their orientation of the γ-phosphoryl group of ATP, immediately adjacent to the recipient Tyr, Ser or Thr residues of the recipient protein [43]. We suggest that, in the MKK3/HvMAPK complex, ATP is hydrolyzed by a single lytic water molecule at its γ-PG-O1G group with the participation of the conserved D211 and D229 residues (fulfilling roles of a catalytic acid or base) [44], with coordinated divalent metal ions (typically Mg^2+^ or Mn^2+^) stabilizing ATP and neighboring residues (Figure 3 and Figure 4).

The released phosphate ion must therefore diffuse along an internal reactant trajectory to reach its target Ser and Thr residues near the DFG HvMAPK activation segment (Figure 3 and Figure 4). At this stage, it is impossible to describe phosphate ion transfer and its trajectory accurately. These processes almost certainly involve local and global conformational changes of HvMKK3 and HvMAPK, and re-distributions of secondary structure elements, and associated electrostatic charges. These adjustments would affect protein structures, as observed in nearly all nucleotide phosphate hydrolases [45], including glucokinases [46]. To further test this phosphate ion diffusion conclusion, gene editing or mutant isolation [47] could be used to generate lines in which the movement of the phosphate ion would be impeded or blocked. However, the potential for the overall 3D structure of the MKK3/HvMAPK complex to be compromised in these genetic variants could place interpretative constraints on the experiments.

Superimposed on these structural limitations are the challenges associated with the MKK3 activity assay, which requires the MKK3 and HvMAPK cDNAs to be expressed in a heterologous system, the isolation of the expressed enzymes and removal of any tags, the phosphorylation and activation of MKK3 by the upstream MAPKKK, and the subsequent incubation of the activated MKK3 with the downstream HvMAPK enzyme and radioactive γ-labelled ATP, in a buffer optimized for pH and ion concentration. The radioactively labelled HvMAPK could be purified by gel electrophoresis, but the amount of enzyme in the gel must be accurately measured for the calculation of reliable specific activities. Furthermore, it would be necessary to confirm that phosphorylation had occurred at the expected activation site residue of the HvMAPK enzyme. Because of the complexity of this assay, painstaking replications and multiple controls would be crucial.

### 2.10. Are Dormancy, Germination, and Carbon/Nitrogen Metabolism Linked?

As mentioned in the Introduction, the two barley genes identified in QTL analyses as regulators of the dormancy/germination transition are MKK3 and AlaAT [4,5]. The AlaAT enzyme catalyzes the reversible transamination reaction:l-alanine + α-ketoglutarate ⇌ pyruvate + l-glutamate,
which links carbohydrate oxidation (pyruvate and α-ketoglutarate) and protein metabolism (alanine and glutamate), and hence C and N metabolism. The AlaAT enzyme has been implicated in nitrogen-use-efficiency and preharvest sprouting [4,47]. Specific alleles of the gene influence the duration of dormancy and the flux through C and N metabolic pathways. For example, of the several AlaAT genes in barley, the AlaAT gene on chromosome 5HL is related to dormancy and enzyme activity [4]. A genetic variant of this gene encodes a Leu to Phe substitution at amino acid residue 214. The L214 allele correlates with longer dormancy, while the F214 allele correlates with shorter dormancy [4]. However, the respective Leu or Phe residues at position 214 are situated some distance from the active site and are believed to be involved in protein–protein interactions, rather than in substrate binding [4].

In addition, the MAPK enzymes RAF14 and RAF79, which are involved in auxin, ABA, and stress signaling in the green alga *Chlamydomonas reinhardtii*, are strongly suppressed by the ammonium ion. MAPK cascades in this alga are activated by nitric oxide and nitrate ions, which result in the regulation of genes that encode nitrate reductase and glutamine synthetase [24]. Nitrate reductase catalyzes the first reaction in nitrate assimilation. Glutamine synthetase is crucial in plant metabolism through its conversion of toxic ammonia into l-glutamine, as follows:l-glutamate + ATP + NH_3_ → l-glutamine + ADP + Pi.(Note that l-glutamate is a reactant in the AlaAT reaction, as shown above.)

Taken together, these data provide strong circumstantial evidence that AlaAT, MKK3, and carbon/nitrogen metabolism, together with a broad range of phytohormone and other signaling molecules, are intricately linked with the length of grain dormancy and the transition to germination. The precise biochemical details of these complex networks are yet to be defined.

## 3. Materials and Methods

### 3.1. Computational Methods

(i) The theoretical 3D models of HvMKK3 and HvMAPK were retrieved from the AlphaFold Protein Structure Database (accession A0A140JZ13) [27] and the SWISS-MODEL Depository (accession A0A8I6YLP6) [48], respectively. The HvMAPK model was constructed by the SWISS-MODEL server using the crystal structure of the kinase domain of the human TRAF2-NCK-interacting protein kinase (PDB accession 5CWZ, chain A) as a template, with 35% sequence identity between the two sequences. The model of HvMKK3 in complex with HvMAPK was calculated using two approaches, based on rigid body docking, without specifying a docked molecule position. The first included HDOCK docking, which is based on a hybrid algorithm of template-based modeling and ab initio free docking of protein–protein and protein–ligand systems to generate docking poses for each model [49]. This process, using 1200 grid spacings of x, y, z translational degrees of freedom and 15,000° angle steps, was repeated three times, predicting 300 models. The second approach utilized the ClusPro docking algorithm [50], where each computed model was based on 70,000 rotations of molecules, with 1000 rotation/translation combinations, followed by subsequent clustering. This docking process was repeated three times. Top models of the HvMKK3/HvMAPK complexes were selected from 300 and 90 models calculated by HDOCK or ClusPro, respectively, based on the positions of dominant phosphorylation motifs in the HvMKK3 and HvMAPK enzymes. Phosphorylation motifs in HvMKK3 and HvMAPK were determined using MusiteDeep [32].

(ii) Models of the two amino acid residues at positions 165 (Glu/E and Gln/Q) and 260 (Asn/N and Thr/T) in HvMKK3 were generated using the mutagenesis wizard in the PyMOL Molecular Graphics System v3.0.7.3 (Schrödinger LLC, New York, NY, USA), selecting the most favorable rotamers, without clashes with neighboring atoms. Selected models were energy minimized using the knowledge-based Yasara2 forcefield (bond distances, planarity of peptide bonds, bond angles, Coulomb terms, dihedral angles, van der Waals forces) [51] combined with the particle-mesh-Ewald energy function for long-range electrostatics at a cut-off of 8.0 Å to obtain smooth electrostatic potentials [52]. During these steps, incorrect covalent geometry and conformational stress were removed by short steepest descent minimization (time 5000 fs, 1 ft time steps, 298 K), followed by simulated annealing with 1 fs time steps, atom velocities scaled down by 0.9 every 10th step, until convergence at 710 steps with energy improvement of >0.05 kJ/mol per atom during 200 steps. The PROCHECK v3.4 program [53] was used to evaluate the geometrical and stereochemical quality of the models, using Ramachandran plots [54]. Structural images were generated in PyMOL v3.1.5.1 (Schrödinger LLC) and visualized in Adobe Illustrator v30.1. The ATP molecule, taken from the human MLK4 kinase domain as AGS (phosphothiophosphoric acid-adenylate ester or ATPγS-ATP analogue; PDB accession 4UYA, chain A), was modified into ATP by replacing the PG-S1G group by the PG-O1G group and energy minimized in YASARA, using the YAMBER forcefield [51].

(iii) For the docking analyses, HDOCK (conducted on 16 July 2025) and ClusPro v2.0 computational tools were chosen because they combine template-based (using known similar structures) and template-free (free docking) methods to improve the prediction accuracy of protein complexes. These tools provide several hundred models, which can be clustered in groups and evaluated. We used the following criteria to select suitable model complexes: (i) The mutual orientation of HvMKK3 and HvMAPK molecules, such that the positions of dominant phosphorylation motifs in the HvMKK3 and HvMAPK enzymes are proximal; (ii) the interface associations between the HvMKK3 and HvMAPK enzymes, which should satisfy stereochemical parameters and favorable contacts within 3.5 Å; (iii) the quality of geometrical and stereochemical parameters of the protein complexes. There were no violations of these parameters in the Ramachandran plots [55], except K180 in HvMAPK, which was also violated in the model template, taken from the Swiss Model database.

(iv) Molecular dynamics (MD) simulations of the HvMKK3/HvMAPK wild-type (with E165 and N260) and Q165 and T260 variant model complexes (three runs for each system with data averaging) were run using CABS-flex v3.0 [37]. The following settings/internal restraints were used: (restraints mode: flexible; protein restraints: flexible 3 3.8 11.5; global sidechain weights 1.0 1.0; percentage of retained restraints 100; gap 3; minimum distance: 3.8 Å; maximum distance: 8.0 Å; sidechain minimum weight 1.0; percentage of key restraints 100, maximum distance 11.5; Cα atoms maximum weight 0.5; side chain maximum weight 0.5; temperature of simulation: 1.40-dimensionless; length: 250 ns or 1,250,000 Monte Carlo steps). These unbiased simulations (without additional restraints) were based on the initial input structures to determine the root mean square fluctuation (RMSF) values (measures of the displacement of the Cα atoms, relative to those of a starting structure, over simulation time). Calculated output trajectories describe structural dynamics based on 5000 models (or 500 models per single all-atom model with the lowest RMSD value to the starting structure) clustered in 10 medoids per class.

### 3.2. Phylogeny of the MKK3 Enzyme

Coding sequences of MKK3 orthologues from the *Triticeae* and related Poaceae species were retrieved from Ensembl Plants (https://plants.ensembl.org/) and manually curated to remove incomplete or low-quality annotations. For barley, we included the canonical MKK3 copy carrying the ancestral N260 residue (HORVU.MOREX.r3.5HG0537440.1) [20]. The final dataset comprised 13 MKK3 CDS from representative *Triticeae* species (Appendix A). The image of the phylogenetic tree (Figure 2) was visualized in Inkscape v1.4.3 (accessed on 21 December 2025).

Protein sequences were aligned using MUSCLE/MAFFT, and the curated protein alignment, together with the corresponding CDS file was converted into a codon alignment using PAL2NAL with default parameters. A maximum-likelihood phylogenetic tree was inferred from the protein alignment using the LG+G substitution model (Geneious v2024.0.4/RAxML settings), and the topology was manually rooted following established *Triticeae* relationships. This tree was used as the fixed phylogeny for all downstream selection analyses.

### 3.3. Ancestral Sequence Reconstruction

Trees containing ancestor and successor sequences were generated using FireProt^ASR^ [34,35] and visualized in FigTree v1.4.4 [55], as described previously [56,57]. The analyses proceeded in two steps with default parameters for automatic evolutionary models and phylogeny settings (bootstraps 50, gap correction 0.5, clustering identity filter 0.9). We used the full-length HvMKK3 or HvMAPK sequences to search for homologous sequences at a 30–90% sequence identity of both proteins to identify ancestral and successor sequences. This step, after homology search and filtering, aggregated datasets of 299 sequences, homologous to HvMKK3 and HvMAPK (step 1), which were used in step 2 to identify 149 ancestral and 150 successor sequences in each case and build phylogenetic trees. Trees, drawn to scale, contained ancestors and successors, with the latter annotated with the National Center for Biotechnology Information (NCBI) accession numbers and, in some instances, with the species of origin. Evolutionary distances were computed using the p-distance method and expressed in units as a residue number difference per site.

### 3.4. Selection Pressure on the N260T Variant

Site-specific selection pressure was assessed using the Datamonkey implementation of HyPhy. For FEL (Fixed Effects Likelihood; http://www.datamonkey.org/), we specified pervasive selection and selected all branches of phylogeny to estimate nonsynonymous (β) and synonymous (α) substitution codon rates. We evaluated whether the residue corresponding to barley N260 could have experienced long-term adaptive evolution. MEME (Mixed Effects Model of Evolution) was run under an episodic selection framework, selecting all branches to detect sites experiencing diversifying selection on one or subsets of lineages. For both FEL and MEME, statistical significance was evaluated using the default likelihood-ratio test and χ^2^ approximation implemented in HyPhy. Summary files containing site-wise estimates of α, β, *dN:dS*, and *p*-values were exported from Datamonkey and processed in R to generate customized visualizations of evolutionary rates along the MKK3 coding sequence, with particular focus on codon 260 (alignment position 262), the position affected by the derived N260T variant in barley.

### 3.5. E165Q and N260T Rotameric Conformers

In PyMol v3.1.5.1, we used the following selection criteria for E165Q and N260T rotameric conformers: (i) backbone-dependent rotamers, (ii) retained H-atoms, (iv) N- and C-caps open, and (iii) without clashes to neighboring atoms (which constituted the 1st stage of selection). The resultant structures with Q165 and T260 residues were minimized, as described above. In the 2nd stage, these variant structures were compared with the 3D models computed in AlphaFold v.3.0 to check the validity of rotameric positions, which agreed with the selection made in PyMol.

### 3.6. Identification of Phosphorylation Sites

We used MusiteDeep [32], a tool based on a Deep-Learning Framework for Protein Post-Translational Modification Site Prediction, to estimate phosphorylation sites in the HvMKK3 and HvMAPK enzymes. We predicted consensus phosphorylation sites on the serine, threonine, and tyrosine (which occurs in less than 1% on tyrosine) residues, together with those containing consensus motifs with a high likelihood of being phosphorylated. The MusiteDeep prediction was corroborated by the NetPhos3.1 analysis [58].

## 4. Summary

Here, we used protein structural modeling of the barley MKK3 enzyme with its downstream substrate enzyme HvMAPK, together with phylogenetic analyses, to define:the number of MAPK genes in the barley genome and their phylogeny;the ancestral origin and phylogeny of the common genetic variants E165Q and T260N, which showed that the wild-type variants were most likely E165 and T260, respectively;the phosphorylation sites of the HvMKK3 enzyme (TFVGTVTY; T245-Y252; phosphorylatable Thr residues underlined) and the HvMAPK enzyme (SLKGTPY; S163-Y169; phosphorylatable residues underlined) and the likely phosphorylated amino acid residues therein;the theoretical molecular docking structure of the HvMKK3/HvMAPK complex and its dynamics;amino acid residues that are likely to be involved in ATP hydrolysis, and the need for the released phosphate group to diffuse through the enzyme to its target phosphorylation loop of the HvMAPK enzyme;the effects of the key genetic variants E165QE and T260N, made possible by a computational model of the enzyme–substrate complex that allowed the rationalization of the effects of electrostatic surface potentials and overall enzyme flexibility provided by the variants on HvMKK3 activity, which occur both at the ATP binding site (E165Q) and at the phosphorylated loops (T260N);a possible explanation of the effects of other common amino acid substitutions (A79V, G350R, N383D) on HvMKK3 activity.

## 5. Future Directions

These data will allow informed approaches to the genetic manipulation of MKK3 structure and function under changing climatic conditions and the attendant benefits associated with optimizing the period of grain dormancy while avoiding undesirable PHS. Alternatively, pre-breeding approaches using the different extant single and multi-copy haplotypes associated with multiple amino acid residue changes could be used to define combinational effects [6].

The MAP kinase cascade, together with the mitogen receptor and the final transcription factor (Figure 1), transfers a mitogen signal from the plasma membrane to the nucleus through a series of phosphorylation reactions. The barley MKK3 enzyme is located in the central section of a MAPK cascade that releases dormancy and allows the barley grain to germinate. The timing of the dormancy to germination transition is crucial for the survival of the young seedling, population survival, crop yields, and industrial utilization of the grain. Key objectives in future work on the dormancy/germination conversion will likely be focused firstly on identifying the mitogen that initiates the cascade and, secondly, on identifying the final transcription factor and the genes that are subsequently transcribed under the influence of that transcription factor.

Strategies to achieve these targets will likely include a combination of genetic, biochemical, structural, and computational biology approaches. One strategy would be to walk step-by-step upstream from MKK3 to the mitogen and step-by-step downstream from MKK3 to the transcription factor. Co-expression and associated network analyses will identify candidate genes and transcription factors, as shown in [59], during the cellularization phase of grain development in barley. Structural biology and computational approaches of the type described in the current work will enable the cascade sequences of interacting upstream and downstream MAP kinase enzymes to be tested and defined, while traditional affinity chromatography methods could be used to confirm the various interactions along the pathway. FIND-IT and CRISPR will be important underlying technologies for the rapid isolation, generation, and testing of candidate genes during this process [46]. Alternatively, one could analyze the multiple existing natural haplotypes in pangenome-informed pre-breeding approaches. A complete definition of the MAPK pathway generated through these types of methods will be crucial for understanding and manipulating the dormancy to germination transition in important crop species.

## Figures and Tables

**Figure 1 ijms-27-00530-f001:**
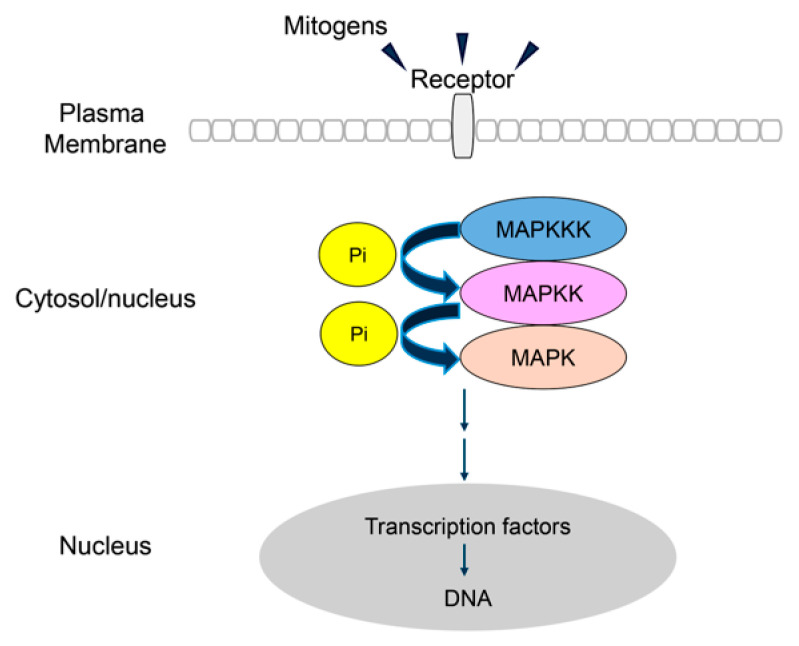
The mitogen-activated kinase cascade. Following the binding of extracellular mitogens to their plasma membrane-bound receptor, a cascade is initiated in which a mitogen-activated kinase kinase kinase (MAPKKK) phosphorylates and activates a mitogen-activated kinase kinase (MAPKK), which in turn phosphorylates and activates a mitogen-activated kinase (MAPK). The MAPKK and MAPK enzymes from barley are referred to in the text as MKK3 and HvMAPK, respectively. The activated MAPK subsequently activates transcription factors in the nucleus. It should be noted that the final MAPK in the cascade can also phosphorylate cytoplasmic enzymes and structural proteins [9]. Re-drawn from Wang and Gou [10].

**Figure 2 ijms-27-00530-f002:**
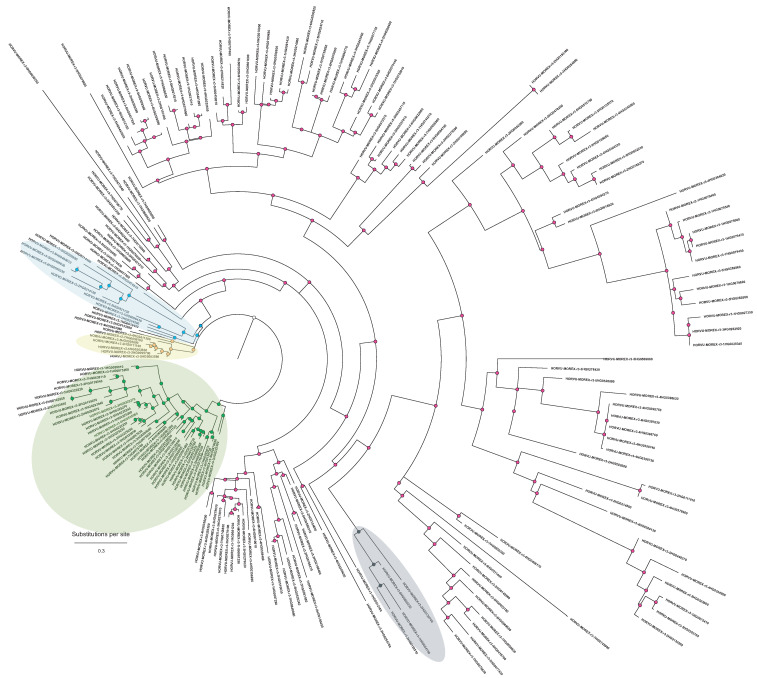
Phylogenetic analysis of the MAPKKKs. Clusters 1 (yellow), 2 (green), and 3 (blue) were composed exclusively of RAF-like MAPKKKs, containing 9, 47, and 10 members, respectively, each forming well-supported and internally coherent RAF sublineages. The largest clade, Cluster 4 (pink), represented a broad macro-expansion that combined 126 RAF-like MAPKKKs with 49 MEKK/ZIK-type MAPKKKs, reflecting ancient diversification and the deep intermixing of kinase lineages when the full PF00069 domain is used for tree reconstruction. In contrast, Cluster 5 (black) contained only MEKK/ZIK-type MAPKKKs (six total), forming a small and well-defined non-RAF lineage consistent with the conserved MEKK and ZIK groups described in other monocots. Visualization of Clusters 1–3 and 5 is enhanced by colored ovals.

**Figure 3 ijms-27-00530-f003:**
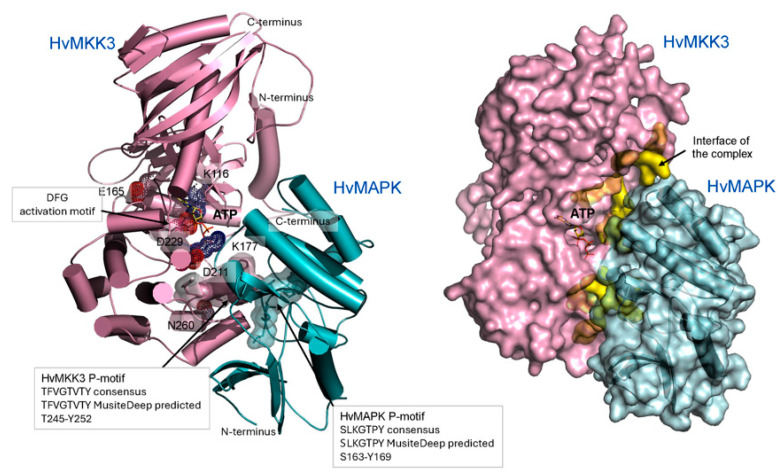
Docking of HvMKK3 and HvMAPK calculated by HDOCK (complex 1). (**Left panel**): Cartoon representation of the 3D HvMKK3 (pink)/HvMAPK (cyan) complex. Catalytic residues K116 and K177 (blue-cpk), and D211 and D229 (red-cpk) are shown in sticks and dots. The activation motif DFG (D229-F230-G231), with D229 contacting ATP, is visualized in translucent pink surface representation. The ATP molecule is shown in yellow-cpk sticks. The respective phosphorylation sites TFVGTVTY (T245-Y252 in HvMKK3) and SLKGTPY (S163-Y169 in HvMAPK), indicated in translucent pink and cyan surface representations, are situated proximally, to allow their direct interaction. (**Right panel**): Surface representation of the HvMKK3 (pink)/HvMAPK (cyan) complex. The cartoon representation of the HvMAPK (cyan) underlying translucent surface is indicated. The ATP molecule is shown in yellow-cpk sticks. Interface residues contributing from each molecule (shown in yellow surface representations) to the complex are at separations between 2.5 Å–2.9 Å. Asp229 is shown as a part of the DFG activation motif. In the HDOCK complex 1 from the ensemble of 100 models, the binding score and RMSD value of the HvMAPK molecule from the initial orientation were −170.19 and 38.57 Å.

**Figure 4 ijms-27-00530-f004:**
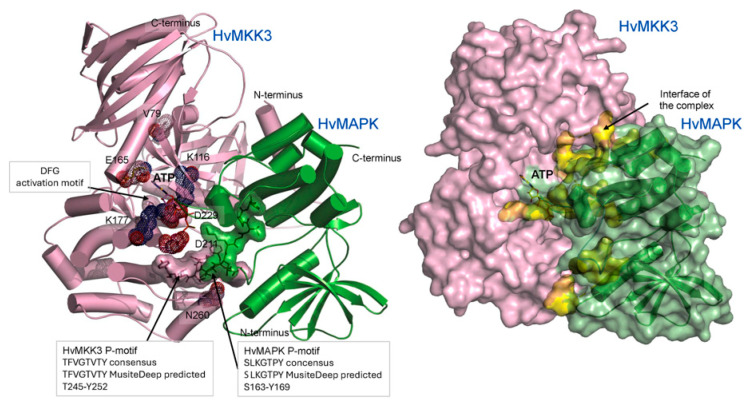
Docking of HvMKK3 and HvMAPK calculated by ClusPro (complex 2). (**Left panel**): Cartoon representation of the 3D HvMKK3 (pink)/HvMAPK (green) complex. Catalytic residues K116 and K177 (blue-cpk), and D211 and D229 (red-cpk) are shown in sticks and dots. The activation motif DFG (D229-F230-G231), with D229 contacting ATP, is visualized in the translucent pink surface representation. The ATP molecule is shown in yellow-cpk sticks. The respective phosphorylation sites TFVGTVTY (T245-Y252) and SLKGTPY (S163-Y169) in HvMKK3 and HvMAPK, indicated in pink and green surface representations, are situated proximally. (**Right panel**): Surface representation of the HvMKK3 (pink)/HvMAPK (green) complex. The cartoon representation of HvMAPK (green) underlying the translucent surface is indicated. Interface residues contributing from each molecule (shown in yellow surface representations) to the complex are at separations between 2.4 Å–3.3 Å. D229 is shown as a part of the DFG activation motif.

**Figure 5 ijms-27-00530-f005:**
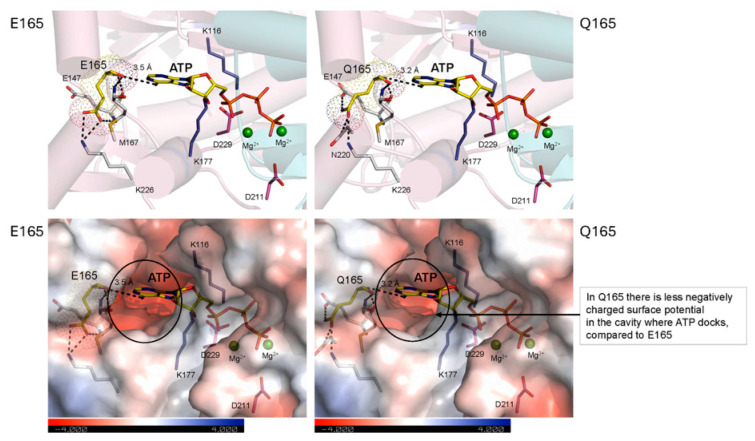
Effects of the E165Q variant. *Top panels*: Cartoon representations of close-up views of E165 wild-type (**left**) and the Q165 mutant (**right**) of the HvMKK3 (pink) components of the HvMKK3/HvMAPK complex, calculated by HDOCK. Diagnostics of AlphaFold v.3.0 models show that E165 is positioned on an α-helix located in the “Very high” probability value (plDDT > 90; the highest accuracy category. Here, pIDDT indicates the “Predicted Local Distance Difference Test”. The root-mean-square deviation value between the two structures is 0.36 Å. Catalytic residues K116 and K177 (blue-cpk), and D211 and D229 (red-cpk) are shown in sticks and dots. E165 (**left panel**) and Q165 (**right panel**) are shown in yellow sticks and dots. Residues interacting with E165 or Q165 are shown in cpk sticks and their separations are indicated in dashed lines. Separations (in Å) between the O atom of E165 or Q165 and AGS (sulfate analogue of ATP) are shown in dashed lines. *Bottom panels*: Close-up views of surface morphologies colored by electrostatic potentials (according to the scales shown on the underlying bar; white, neutral; blue, +4 kTe^−1^; red, −4 kTe^−1^) of E165 wild-type (**left**) and the Q165 mutant (**right**) of the HvMKK3 components of complexes, calculated by HDOCK. Catalytic residues K116 and K177 (blue-cpk), and D211 and D229 (red-cpk) are shown in sticks. E165 and Q165 are shown in yellow-cpk sticks and dots with neighboring E147, M167 and K226 for E165 (separations in dashed lines between 2.8 Å–3.1 Å), and E147, M167, N220, and K226 (separations in dashed lines between 2.8 Å–3.2 Å). Separations indicated in Å between E165 or Q165 and ATP are in dashed lines. Circles indicate differences in electrostatic potentials around E165 and Q165 in proteins. N229 is a component of the DFG activation site.

**Figure 6 ijms-27-00530-f006:**
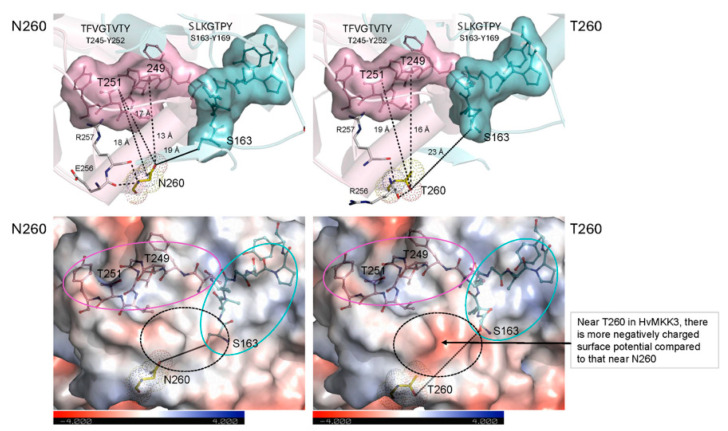
Effects of the N260T variant. *Top panels*: Cartoon representations of close-up views of T260 wild-type (**left**) and the N260 variant (**right**) of the HvMKK3 (pink) components of the HvMKK3/HvMAPK complex, calculated by Clus-Pro. The diagnostics of residues in AlphaFold v.3.0 models show that N260 is positioned on a short loop, located to the “Confident category (90 > plDDT > 70). The root-mean-square deviation value between two structures is 0.40 Å. T260 (**left panel**) and N260 (**right panel**) are shown in yellow-cpk sticks and dots. The respective phosphorylation sites TFVGTVTY (consensus) and SLKGTPY (predicted by MusiteDeep) in HvMKK3 and HvMAPK, indicated in pink and cyan surface representations (also marked above images), are situated proximally. Separations between N260 or T260 and phosphorylatable residues in TFVGTVTY and SLKGTPY motifs are in dashed lines. *Bottom panels*: Close-up surface morphology views, colored by electrostatic potentials (according to the scales shown on the underlying bar; white, neutral; blue, +4 kTe^−1^; red, −4 kTe^−1^) of T260 wild-type (**left**) and the N260 variant (**right**) of HvMKK3 components of the complex. The respective phosphorylation sites TFVGTVTY (consensus) and SLKGTPY (predicted by MusiteDeep) in HvMKK3 and HvMAPK, indicated in pink-cpk and cyan-cpk lines and spheres, and circles, are situated proximally.

**Figure 7 ijms-27-00530-f007:**
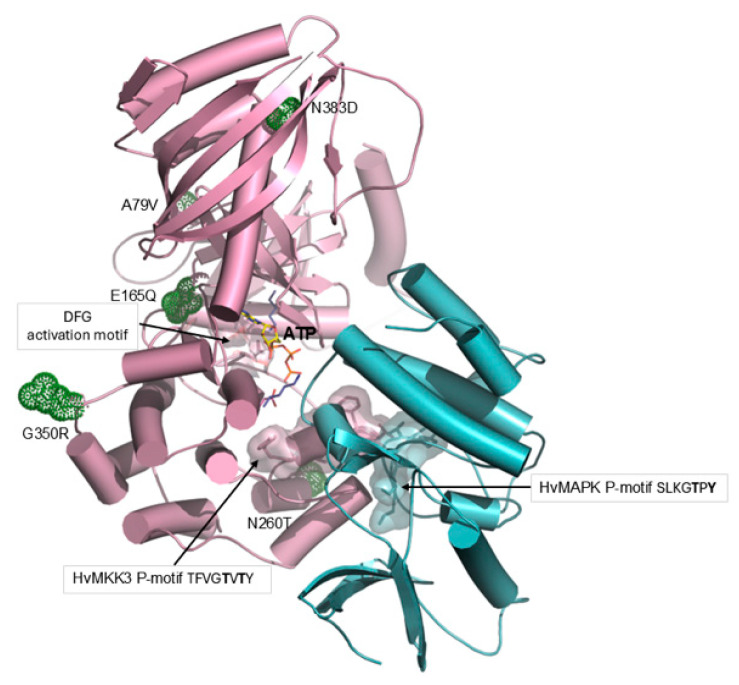
Positions of common genetic variants of the barley MKK3 enzyme. Second-named amino acid residues A79V, E165Q, N260T, G350R, and N383D are shown in green-cpk sticks and dots.

**Table 1 ijms-27-00530-t001:** Summary of common genetic variants of the MKK3 enzyme from barley ^a^.

Genetic Variants	MKK3 Activity	Dormancy	Risk of PHS
E165 Q165	lower higher	longer shorter	lower higher
N260 T260	higher lower	shorter longer	higher lower
A79 V79	lower higher	longer shorter	lower higher
G350 R350	not known (slightly lower mRNA) not known (slightly higher mRNA)	not known not known	not known not known
N383 D383	higher (slightly higher mRNA) lower (slightly lower mRNA)	not known not known	not known not known

^a^ Data from [5,6]. Note that these assignments should be viewed with caution, given the complexities placed on their interpretation by the widespread variation in *MKK3* gene copy number, by the occurrence of complex mixtures of haplotypes, and through alternative splicing.

## Data Availability

The data presented in this study are available upon request from the corresponding author.

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
