# Peer review of "Dormancy Versus Germination: 3D Protein Modeling and Evolutionary Analyses Define the Roles of Genetic Variants in the Barley MKK3 Enzyme"

_ijms, 2026, doi:10.3390/ijms27010530_

Round 1

Reviewer 1 Report

Comments and Suggestions for Authors

In this work, the authors reported that mechanism of ATP hydrolysis and the effects of common genetic variants on MKK3 activity, thereby influencing the release from dormancy. This work is very meaningful and clear. However, there are several concerns as follows.

  1. The ATP binding mechanism requires supplementary energy data support. The manuscript speculates that ATP diffuses into the active site after the formation of the complex, but lacks quantitative energy evidence. It is suggested to add MM/PBSA free energy calculationto compare the differences in ATP binding affinity between the E165Q/N260T variant and the wild type, verify the hypothesis that changes in electrostatic potential affect ATP binding, and enhance the persuasiveness of the conclusion.

  1. The evolutionary analysis of the N260T variant needs to be refined. The manuscript mentions that N260T might have been produced by S260 through a single nucleotide mutation, but it does not provide evidence of evolutionary selection pressure. It is suggested to supplement the calculation of the dN/dS ratio or the selection pressure analysis of the ancestral sequence to clarify the evolutionary origin of this variant (such as whether it is an adaptive mutation) and its selection preferences in different environments, and to improve the explanation of the evolutionary mechanism.

  1. The materials and methods need to be supplemented in more detail. For example, in sections 2.2 and 2.3.

Author Response

Responses to reviewers’ comments and questions

Reviewer1          

Comments and Suggestions for Authors

In this work, the authors reported that mechanism of ATP hydrolysis and the effects of common genetic variants on MKK3 activity, thereby influencing the release from dormancy. This work is very meaningful and clear. However, there are several concerns as follows.

  1. The ATP binding mechanism requires supplementary energy data support. The manuscript speculates that ATP diffuses into the active site after the formation of the complex but lacks quantitative energy evidence. It is suggested to add MM/PBSA free energy calculation to compare the differences in ATP binding affinity between the E165Q/N260T variant and the wild type, verify the hypothesis that changes in electrostatic potential affect ATP binding, and enhance the persuasiveness of the conclusion.

Authors’ response:

We acknowledge that this comment is fair. We have not explored the diffusion process of ATP in the current work, and our suggestions regarding this process were purely hypothetical. Studies using MM/PBSA free energy calculations, Reviewer 1 suggests, are beyond the scope of this work. Alternative approaches could also be used, as we demonstrated in our studies of reactant trajectories in an exo-hydrolytic enzyme (Nature Communications 2019, 2022, and Communications Biology 2025). However, both approaches are computationally costly and demand significant time, manpower, and resources. To resolve this situation, we have toned down our statements regarding ATP diffusion and movements.

Nevertheless, in response to the comment, we provide new molecular dynamics simulation data, using a coarse-grained approach with all-atom details (as implemented in CABS-flex 3.0) on model complexes. These data do not substitute for MM/PBSA free energy calculations, but nevertheless, provide further details on the structural dynamics of protein complexes. These data are presented in the new Supplementary Figure S5 and discussed in section 3.5.

  1. The evolutionary analysis of the N260T variant needs to be refined. The manuscript mentions that N260T might have been produced by S260 through a single nucleotide mutation, but it does not provide evidence of evolutionary selection pressure. It is suggested to supplement the calculation of the dN/dS ratio or the selection pressure analysis of the ancestral sequence to clarify the evolutionary origin of this variant (such as whether it is an adaptive mutation) and its selection preferences in different environments, and to improve the explanation of the evolutionary mechanism.

Authors’ response:

We have now performed a dedicated codon-based selection analysis, with calculated dN:dS ratios, of MKK3 orthologues across the Triticeae to clarify the evolutionary context of the N260 site. The data are consistent with N260 being a recent, domestication-associated mutation that arose within a constrained site and has been retained under human-driven selection. The methods used are described in a new section 2.4, and the data are discussed in detail at the end of section 3.1.

  1. The materials and methods need to be supplemented in more detail. For example, in sections 2.2 and 2.3.

                Authors’ response:

These details have been provided, and the sections have been significantly extended. These descriptions are included in the Materials and Methods sections 2.3 and 2.6 (please note new numbering, as Materials and Methods were re-arranged).

Reviewer 2 Report

Comments and Suggestions for Authors

Dear Authors,

MAPK cascades have been related to a wide range of processes, including responses to biotic and abiotic stresses, development, growth, and hormonal signaling. In this paper, the authors study one of these, MKK3, which is involved in dormancy in barley. I believe that the article, in its current form, has a number of shortcomings that necessarily need to be addressed, I also think it contains significant gaps in information that the authors should address. I have provided recommendations on how to do so.

Majors:

-The field of MAPKs is very broad, and this is one of the aspects in which the authors fall short; the introduction and discussion is too specific to a single MAPK and does not extrapolate to others in the family and other organisms.

-In my view, the introduction does not correctly explain the role of MAPKs; it could be improved by providing more detail on the many aspects in which they are involved. For example, the different types of stress in which they are involved (oxidative, carbon, nitrogen,…).

-The introduction should reflect the number of MAPKs in barley for each group and compare them with other model organisms, such as Arabidopsis, humans, or Chlamydomonas.

-A phylogenetic tree of barley MAPKs would be highly necessary

-Explain in more detail why HDOCK and ClusPro were specifically selected, and how their results are compared (i.e., the objective criteria used to select the best models beyond the proximity of phosphorylatable motifs).

-Provide a stronger justification for the specific scoring parameters and docking quality metrics used (e.g., energies, scores, interface areas) to support that the complexes presented are indeed the most plausible ones.

-For the variant modeling, mutagenesis in PyMOL and minimization are described, but it would be useful to indicate whether the relative stability of the variants was evaluated, and not only their local geometry, please.

-At least in my file, the Figures does not have sufficient resolution.

-Provide a more detailed explanation of how multiple rotamers or conformers were analyzed, and whether the described effects are robust to this variability.

-Ancestral reconstruction using FireProtASR is mentioned, but it is described in very general terms; it would be advisable to add information about the input alignment (size, taxonomic coverage) and the evolutionary model used. Please improve

-Specify the posterior probability values of the key residues (E165Q, N260T) in the relevant ancestral nodes, so that the inference about the “wild type” is quantitatively more robust

-In the case of N260T, elaborate further on how the changes in electrostatic potential could affect the transition-state energy or the frequency of “competent” conformations for phosphorylation.

-In my view, the discussion is too structural and does not correctly relate to previous studies. How is dormancy connected to basic metabolism, such as nitrogen or carbon? Is the relationship of this MKK3 with nitrogen metabolism through dormancy, as seen with RAF14 and RAF79, plausible? please discuss

-The section in which it is proposed that inorganic phosphate diffuses internally after ATP hydrolysis to reach the SLKGTPY motif of HvMAPK is interesting but highly speculative. It would be important to clearly label it as a “working model” and to discuss other, more conventional alternatives in greater Depth.

- Nitrogen can act as a signaling molecule that has been described as involved in dormancy, and MAPKs involved in nitrogen metabolism have been reported; please discuss this

-In my view, the connection between the modeling results and the previous functional data could be made more explicit and critical. For example, it would be helpful to clearly indicate which concrete predictions arise from the model that could be experimentally tested, such as additional mutants or activity assays using ATP analogs.

Author Response

Responses to reviewers’ comments and questions

Reviewer-2         

Dear Authors,

MAPK cascades have been related to a wide range of processes, including responses to biotic and abiotic stresses, development, growth, and hormonal signaling. In this paper, the authors study one of these, MKK3, which is involved in dormancy in barley. I believe that the article, in its current form, has a number of shortcomings that necessarily need to be addressed, I also think it contains significant gaps in information that the authors should address. I have provided recommendations on how to do so.

Majors:

-The field of MAPKs is very broad, and this is one of the aspects in which the authors fall short; the introduction and discussion is too specific to a single MAPK and does not extrapolate to others in the family and other organisms.

-In my view, the introduction does not correctly explain the role of MAPKs; it could be improved by providing more detail on the many aspects in which they are involved. For example, the different types of stress in which they are involved (oxidative, carbon, nitrogen…).

Authors’ response:          

The MAPK field is certainly very broad, but we have now addressed this suggestion to broaden the background information on plant MAPKs. We have extended a paragraph in the Introduction to summarize the range of external signals (including stress) that are transmitted through the MAPK cascade and the developmental changes that are induced. More details of the responses are provided in the new section 3.9. The legend of Figure 1 has also been revised to reflect the importance of signaling responses associated with cytoplasmic proteins.   

 The introduction should reflect the number of MAPKs in barley for each group and compare them with other model organisms, such as Arabidopsis, humans, or Chlamydomonas.

-A phylogenetic tree of barley MAPKs would be highly necessary.

Author’s response:          

These MAPK gene numbers are now compared in the Introduction. A phylogenetic tree is also included in Figure 2, the methods used are described in section 2.1 and the results discussed in section 3.1. We have also modified the title of the paper to reflect this additional information. 

Explain in more detail why HDOCK and ClusPro were specifically selected, and how their results are compared (i.e., the objective criteria used to select the best models beyond the proximity of phosphorylatable motifs).

Author’s response:          

HDOCK and ClusPro computational tools were chosen because they combine template-based (using known similar structures) and template-free (free docking) methods to improve the prediction accuracy of protein complexes. The details of these methods and the criteria for selection of the suitable model complexes are now included in section 2.1 (paragraph c).

-Provide a stronger justification for the specific scoring parameters and docking quality metrics used (e.g., energies, scores, interface areas) to support that the complexes presented are indeed the most plausible ones.

Authors’ response:

The selection of suitable model complexes was primarily based on the mutual orientation of HvMKK3 and HvMAPK molecules, to support the working hypothesis that HvMKK3 phosphorylates its downstream HVMAPK substrate. We have chosen two independent docking approaches, which use completely different algorithms. But in the end, we reached the consensus.

The ClusPro does not score resultant models and relies on the evaluation of geometrical and stereochemical parameters instead. In fact, ClusPro developers strongly encourage the user not to judge models based on scores because that is not what the scoring function in ClusPro was designed for. As for the geometrical and stereochemical parameters of the ClusPro, complex 2 (Figure 4), these values are correct. We have also evaluated in detail mutual positions of both molecules in complexes. We have included new data of this comparison for both complexes in the Cartesian space. The RMSD values, using two independent procedures, indicate that the two models could represent intermediates during the docking of HvMAPK into HvMKK3. Finally, we have evaluated interface profiles of complexes, which are detailed in legends to Figure 3 (calculated by HDOCK; complex 1) and Figure 4 (calculated by ClusPro; complex 2).

These points are now explained in sections 2.1 (paragraph c) and 3.4.

-For the variant modeling, mutagenesis in PyMOL and minimization are described, but it would be useful to indicate whether the relative stability of the variants was evaluated, and not only their local geometry, please.

Authors’ response

The answer has been provided in the previous paragraph. However, to add to this point, and as described in the response to Reviewer 1, we now provide molecular dynamics simulation data of model complexes in the new Supplementary Figure S5. We also provide a detailed description of this approach and the interpretation of data in the Results and Discussion section. We have identified additional structural factors controlling the function of the wild-type variant complexes. Here, alongside electrostatic influence and consequences, there are inherent structural constraints that control the dynamics of the wild-type and variant HvMKK3/HvMPAK complexes.

At least in my file, the Figures does not have sufficient resolution.

Author’s response:

We have now provided new Figures, redrawn in Adobe Illustrator, and Inkscape at high resolution. This is relevant specifically to Figure 1 (at 300 DPI), where the reader can zoom-in to inspect individual labels if needed at high resolution.

Provide a more detailed explanation of how multiple rotamers or conformers were analyzed, and whether the described effects are robust to this variability.

Author’s response:

We have introduced a new section to Materials and Methods (2.5. E165Q and N260T rotameric conformers), where we described in detail the method.

In PyMol Molecular Graphics System v3.1.5.1, we used the following selection criteria for E165Q and N260T rotameric conformers: (i) backbone-dependent rotamers, (ii) retained H-atoms, (iv) N- and C-caps open, and (iii) without clashes to neighbouring atoms (which constituted the 1st stage of selection). The resultant structures with E165Q and N260 mutated residues were minimized, as described. In the second stage these mutated structures were compared with 3D models generated by AlphaFold v.3.0 (we computed E165, E165Q and N260T structures) to check the validity of rotameric positions, which agreed with the selection of rotamers via PyMol Molecular Graphics System v3.1.5.1. This information is now included in Section 2.5.

As for the diagnostics of E165 and N260 residues in AlphaFold v.3.0 models, E165 is positioned on an α-helix located in the “Very high” probability value (plDDT > 90; the highest accuracy category), and N260 positioned on a short loop, located to the “Confident category (90 > plDDT > 70). Here, pIDDT indicates the “Predicted Local Distance Difference Test”. These confidence levels are now included in the legends of Figures 6 and 7.

Ancestral reconstruction using FireProtASR is mentioned, but it is described in very general terms; it would be advisable to add information about the input alignment (size, taxonomic coverage) and the evolutionary model used. Please improve

Authors’ response:

The detailed description of ancestral sequence reconstruction using FireProtASR has been provided above (in response to Reviewer 1), including all selected parameters. No taxonomic coverage selection could be selected for analyses using FireProtASR.

-Specify the posterior probability values of the key residues (E165Q, N260T) in the relevant ancestral nodes, so that the inference about the “wild type” is quantitatively more robust

Author’s response:

We have provided posterior probability values for successors in three clusters, related to ancestor 229 (from which HvMKK3 originates), and ancestors 200, and 299. Supplementary Figure S3 and its legend have been modified accordingly to define these parameters. We have also modified legends to Supplementary Figures S4 and S5 to provide more detailed and clearer descriptions.

-In the case of N260T, elaborate further on how the changes in electrostatic potential could affect the transition-state energy or the frequency of “competent” conformations for phosphorylation.

Authors’ response:

We have now commented on electrostatic potentials in section 3.7 of the manuscript and supported our conclusions with a new Supplementary Figure S5, panels F. We have also cautioned against over-interpretation of protein complex models generated by docking (and not 3D complexes, backed by electron densities or cryogenic-EM maps).

In my view, the discussion is too structural and does not correctly relate to previous studies. How is dormancy connected to basic metabolism, such as nitrogen or carbon? Is the relationship of this MKK3 with nitrogen metabolism through dormancy, as seen with RAF14 and RAF79, plausible? please discuss

Author’s response:

We have now added a new section 3.9, entitled Are dormancy, germination and carbon/nitrogen metabolism linked? We cite evidence from plants and Chlamydomonas that suggests how MKK3, carbon and nitrogen metabolism and AlaAT might be linked with dormancy and germination.   

-The section in which it is proposed that inorganic phosphate diffuses internally after ATP hydrolysis to reach the SLKGTPY motif of HvMAPK is interesting but highly speculative. It would be important to clearly label it as a “working model” and to discuss other, more conventional alternatives in greater Depth.

Authors’ response: 

We have re-labelled the heading of section 3.8 “Mechanism of MKK3 Action: a working model”. In our defence, we did state that ‘At this stage, it is impossible to describe phosphate ion transfer and its trajectory accurately’ but acknowledge that our conclusions regarding the enzymic mechanism indeed constitute a working model. We have also added a sentence on the more conventional reaction mechanisms of plant protein kinases in this section.

- Nitrogen can act as a signaling molecule that has been described as involved in dormancy, and MAPKs involved in nitrogen metabolism have been reported; please discuss this

Authors’ response:

As noted above, the new section 3.9 describes the signaling molecules and the cellular responses that are linked with N metabolism.

In my view, the connection between the modeling results and the previous functional data could be made more explicit and critical. For example, it would be helpful to clearly indicate which concrete predictions arise from the model that could be experimentally tested, such as additional mutants or activity assays using ATP analogs.

Authors’ response:

Of the conclusions and predictions listed in the Summary section 4, mutants could be generated to test the proposed phosphorylated residues in both HvMKK3 and HvMAPK, and the amino acid residues that are likely to be involved in ATP hydrolysis. It might also be possible to generate mutants to test our conclusion that the released phosphate ion diffuses to the phosphorylated amino acid residues on the HvMAPK enzyme, as is now outlined in the final paragraph of the revised section 3.8. Our major concern with these approaches, which could take many months to complete, is that the genetic variants generated might alter 3D conformations of either enzyme or of the HvMKK3/HvMAPK complex and hence impose interpretative problems on activity measurements.

In addition, further experimental testing of the predicted phosphorylation sites would be limited by the inherently complexity and associated unreliability of the in vitro MKK3 assay. We have commented on this at the end of section 3.8. 

Round 2

Reviewer 2 Report

Comments and Suggestions for Authors

Dear Authors,,

I believe the authors have adequately addressed all of my comments and suggestions, and I accept the paper in its current version.